# Regulatory RNA Networks in Ovarian Follicular Cysts in Dairy Cows: Implications for Human Polycystic Ovary Syndrome

**DOI:** 10.3390/genes16070791

**Published:** 2025-06-30

**Authors:** Ramanathan Kasimanickam, Vanmathy Kasimanickam, Joao Ferreira, John Kastelic, Fabiana de Souza

**Affiliations:** 1College of Veterinary Medicine, Washington State University, Pullman, WA 99164, USA; vkasiman@wsu.edu; 2School of Veterinary Medicine and Animal Science, São Paulo State University-UNESP, Botucatu 18618-681, São Paulo, Brazil; joao.cp.ferreira@unesp.br (J.F.); fabiana.f.souza@unesp.br (F.d.S.); 3Faculty of Veterinary Medicine, University of Calgary, Calgary, AB T2N 4Z6, Canada; jpkastel@ucalgary.ca

**Keywords:** cystic ovaries, cystic ovarian disease, microRNA dysregulation, transcriptomic network, cattle, women

## Abstract

**Background/Objectives**: Ovarian follicular cysts (OFCs) in dairy cows represent a significant cause of infertility and share striking similarities with polycystic ovary syndrome (PCOS) in women. This study aimed to elucidate the molecular mechanisms underlying OFCs and their relevance to PCOS by profiling differentially expressed (DE) microRNAs (miRNAs) and constructing integrative RNA interaction networks. **Methods**: Expression analysis of 84 bovine miRNAs was conducted in antral follicular fluid from normal and cystic follicles using miScript PCR arrays. Bioinformatic tools including miRBase, miRNet, and STRING were employed to predict miRNA targets, construct protein–protein interaction networks, and perform gene ontology and KEGG pathway enrichment. Network analyses integrated miRNAs with coding (mRNAs) and non-coding RNAs (circRNAs, lncRNAs, snRNAs). **Results**: Seventeen miRNAs were significantly dysregulated in OFCs, including bta-miR-18a, bta-miR-30e-5p, and bta-miR-15b-5p, which were associated with follicular arrest, insulin resistance, and impaired steroidogenesis. Upregulated miRNAs such as bta-miR-132 and bta-miR-145 correlated with inflammation, oxidative stress, and intrafollicular androgen excess. Key regulatory lncRNAs such as Nuclear Enriched Abundant Transcript 1 (NEAT1), Potassium Voltage-Gated Channel Subfamily Q Member 1 Opposite Strand/Antisense Transcript 1 (KCNQ1OT1), Taurine-Upregulated 1 (TUG1), and X Inactive Specific Transcript (XIST), as well as circRNA/pseudogene hubs, were identified, targeting pathways involved in metabolism, inflammation, steroidogenesis, cell cycle, and apoptosis. **Conclusions**: The observed transcriptomic changes mirror core features of human PCOS, supporting the use of bovine OFCs as a comparative model. These findings provide novel insights into the regulatory RNA networks driving ovarian dysfunction and suggest potential biomarkers and therapeutic targets for reproductive disorders. This network-based approach enhances our understanding of the complex transcriptomic landscape associated with follicular pathologies in both cattle and women.

## 1. Introduction

Cystic ovarian disease (COD) is a major cause of reproductive failure in dairy cattle. Ovarian follicular cysts (OFCs) are anovulatory follicular structures, >2.5 cm in diameter, that persist for at least 10 days in the absence of a corpus luteum. The incidence of COD ranges from 2.7 to 30% in lactating dairy cows [1,2,3]. These cysts cause substantial reproductive problems, including reduced conception rates, prolonged intervals to become pregnant, and higher culling rates due to infertility, resulting in considerable economic losses [4].

The development of OFCs in dairy cows is attributed to endocrine disturbances within the hypothalamic–pituitary–gonadal axis (HPGA), triggered by endogenous and exogenous factors [5,6,7,8]. The widely accepted hypothesis is that alterations in luteinizing hormone (LH) release from the anterior pituitary lead to an absence, reduced magnitude, or improper timing of the preovulatory LH surge during maturation of dominant ovarian follicles [9,10,11]. Additionally, progesterone (P4) plays a key role in ovarian cyst formation, with a strong association between intermediate P4 concentrations (0.1 to 1.0 ng/mL) in peripheral blood and the occurrence of OFCs. Most cysts are accompanied by decreased P4 concentrations (<0.1 ng/mL), which further promote cyst development [12]. Based on molecular analyses, bovine ovarian cysts have disrupted hormone receptor patterns, particularly those associated with receptors for follicle-stimulating hormone (*FSHR*), progesterone (*PGR*), LH/choriogonadotropin (*LHCGR*), and estrogen (*ESR*) [4,13,14].

Reproductive success is a key component of lifetime efficiency in dairy cows, whereas negative energy balance (NEB) is an important contributor to reproductive disorders [15]. This imbalance occurs when dietary energy intake is insufficient to meet the demands of high milk production during early postpartum lactation. Furthermore, NEB is often accompanied by hormonal and metabolic changes that disrupt normal ovarian function [16,17] and NEB can also promote the formation of OFCs [8]. During NEB, serum concentrations of insulin-like growth factor 1 (IGF-1), insulin [16], and leptin (LEP) [18,19] decrease. Low systemic concentrations of IGF-1 in the early postpartum period can result in ovulation failure and the development of OFCs [20]. Similarly, reduced serum insulin concentrations during early postpartum may contribute to ovarian dysfunction, including cyst formation [21]. Spicer [22] suggested that when leptin concentrations exceed a threshold, it can trigger hypothalamic–pituitary gonadotropin secretion. In moderate- to high-leptin conditions, similar to obesity, leptin regulates ovarian steroidogenesis [22]. Previous studies have highlighted that molecular mechanisms underlying the formation of OFCs in dairy cows are complex. However, hormone metabolic profiles and RNA sequencing have not been widely used to analyze large-scale gene expression patterns that may explain molecular mechanisms underlying OFC formation.

Based on hormone and gene expression, antral fluids of OFCs were characterized by significantly lower concentrations of E2, insulin, IGF-1, and leptin, but elevated ACTH and ghrelin compared to antral fluids of normal ovarian follicles. The mRNA expression of the corresponding receptors in *theca interna* cells, including *PGR*, *ESR1*, *ESR2*, *IGF1R*, *LEPR*, *Insulin-Like Growth Factor-Binding Protein 6 (IGFBP6)*, and *Growth Hormone Secretagogue Receptor (GHSR)*, was also significantly altered [23]. Furthermore, RNA sequencing identified 2514 differentially expressed genes between normal ovarian follicles and OFCs. Kyoto Encyclopedia of Genes and Genomes (KEGG) pathway analysis connected the ovarian steroidogenesis pathway, particularly the Steroidogenic Acute Regulatory Protein (*STAR*), 3 beta-hydroxysteroid dehydrogenase (*3β-HSD*), Cytochrome P450 Family 11 Subfamily A Member 1 (*CYP11A1*), and Cytochrome P450 Family 17 Subfamily A Member 1 (*CYP17A*1) genes, to the formation of OFCs.

MicroRNAs (miRNAs) are small, non-coding RNAs in various tissues [24,25,26] and biofluids [27,28,29], where they regulate the expression of target genes, at both post-transcriptional and translational levels. These extracellular miRNAs can be detected in several body fluids, including blood serum and plasma, urine, saliva, semen, and milk [30,31]. Although miRNAs have promise as biomarkers for various developmental stages and diseases, the origins of extracellular miRNAs and their mechanisms of communication between the cells of origin and target cells remain poorly understood. The complexities of isolating and characterizing RNA present a challenge. Recently, we explored the expression patterns of mature miRNAs in serum, sperm, and seminal plasma [28,32,33].

Many features of OFCs in cows parallel those of polycystic ovary syndrome (PCOS) in women, a complex endocrine disorder with anovulation, hyperandrogenism, and metabolic disturbances. Therefore, dairy cows have potential as a comparative model for PCOS. Recent transcriptomic advancements highlighted the roles of non-coding RNAs (ncRNAs), including microRNAs (miRNAs), long non-coding RNAs (lncRNAs), small nuclear RNAs (SnRNAs), circular RNAs (circRNAs), and pseudogenes, in orchestrating gene regulatory networks underlying both conditions.

The present study aimed to investigate mechanisms underlying OFCs by analyzing differentially expressed miRNAs, performing in silico analyses of regulatory interactions among coding and non-coding RNAs, elucidating the molecular basis of OFCs in dairy cows and exploring translational insights for PCOS.

## 2. Materials and Methods

### 2.1. Ethical Statement

According to Washington State University Institutional Animal Care and Use Committee (IACUC) Policy #21, this study was deemed exempt. All procedures were part of routine clinical reproductive management. Experimental groups were defined retrospectively based on the presence or absence of ovarian follicular cysts (OFCs).

### 2.2. Animals

Ten lactating Holstein cows in their second or third lactation, with no history of postpartum uterine disease, were included. The cows were fed a total mixed ration twice daily to meet or exceed nutritional requirements for lactating Holsteins weighing approximately 685 kg (range: 625–750 kg) and producing 27–35 kg of 3.5% fat-corrected milk. Based on reproductive examinations via transrectal palpation and ultrasonography (linear-array transducer; SonoScape S8, Universal Imaging Inc., Bedford Hills, NY, USA), cows were retrospectively assigned to one of two groups: normal (*n* = 5), without OFCs, and cystic (*n* = 5), diagnosed with OFCs. The mean (±SD) days postpartum at enrollment were 52 ± 4.5 (range: 47–63) for the normal group and 54 ± 5.6 (range: 50–61) for the cystic group.

Cows in the normal group underwent ovum pickup (OPU) at random stages of the estrous cycle. Mature ovarian follicles were identified using transrectal palpation and ultrasonography, with antral diameters ranging from 15 to 20 mm. Cows in the cystic group were diagnosed during postpartum reproductive exams. Follicular cysts were defined as anovulatory, thin-walled, fluid-filled structures (≥25 mm) persisting for at least 10 days in the absence of a corpus luteum, with antral diameters ranging from 26 to 33 mm.

### 2.3. Sample Collection

Cows were restrained in an adjustable squeeze chute, and the rectum was emptied by back-raking to prevent air aspiration. The vulva and perineal area were washed with water and dried, and the tail was secured to the cow’s neck. Caudal epidural anesthesia was administered using 3–5 mL of 2% lidocaine hydrochloride (Vet One, Boise, ID, USA) injected into the sacrococcygeal epidural space to reduce abdominal straining and facilitate transrectal manipulation. A transvaginal ultrasound transducer, cleaned and fitted to a convex transducer holder (Minitube USA Inc., Verona, WI, USA), was lubricated and inserted into the anterior vagina. The transducer was positioned laterally to the external os of the cervix (fornix vagina), and the ovary was gently manipulated against the transducer to visualize the target follicle or OFC.

Once aligned, a long-beveled aspiration needle (18 G X 3”, Minitube USA Inc.) was used to puncture the vaginal wall and the outer layers of the follicle or OFC. Antral fluid was aspirated using a vacuum pump (Cook^®^ Vacuum Pump, Cook Medical LLC, Bloomington, IN, USA) set to 70 ± 5 mm Hg. One end of the tube was connected to the aspiration needle, and the other end to the vacuum pump, with fluid accumulating in 50 mL collection tubes. Retrieved oocytes of suitable quality were transported in holding media to the laboratory for in vitro embryo production. Antral fluid samples from each mature follicle or OFC (5 cows per group) were placed in labeled cryovials, snap-frozen, stored in liquid nitrogen (−196 °C), and transported to the laboratory.

### 2.4. Follicular Fluid Mature miRNA Profiling Using Real-Time PCR

Small RNAs were separately purified (for each cow) from the antral fluid of individual mature follicles or OFCs, using the miRNeasy serum/plasma kit (Qiagen Inc., Valencia, CA, USA), which uses phenol/guanidine lysis and silica-membrane column isolation of cell-free small RNAs [28,32,33]. After adding QIAzol reagent, spike-in controls, and chloroform to thawed samples, the mixture was shaken, incubated, and centrifuged. The aqueous phase was collected, mixed with ethanol, and passed through an RNeasy MinElute column. The RNA was bound to the membrane, washed, and eluted with RNase-free water. For complementary DNA synthesis, total RNA was reverse-transcribed using a miScript II RT kit (Qiagen Inc.), following incubation at 37 °C for 60 min and 95 °C for 5 min. The resulting cDNA was stored at −20 °C.

For miRNA profiling, real-time PCR was performed using miScript miRNA PCR arrays (Appendix A) with the miScript SYBR Green PCR Kit (Qiagen Inc.), targeting 84 prioritized bovine mature miRNAs from miRBase Version 20 (accessed on 22 April 2020) [28]. Controls were included for normalization and performance assessment. The miRNAs were amplified using a StepOnePlus cycler (Applied Biosystems, Foster City, CA, USA), and cycle conditions included denaturation, annealing, and extension steps. To analyze differentially expressed (DE) miRNAs, raw CT data were uploaded to a web-based platform for quality control and statistical analysis. The CT values were normalized to internal controls, and ΔCT, 2^−ΔCT, fold change, and *p*-values were calculated using Student’s *t*-test.

### 2.5. Bioinformatics Analyses

#### 2.5.1. Conserved Nucleotide Sequences

Differentially expressed miRNA (DE-miRNA) sequences from cattle and humans were retrieved from miRbase v22.1 (www.mirbase.org) (accessed on 13 February 2025) and compared to assess sequence similarity [34,35].

#### 2.5.2. Identification of Target Genes

Target genes of the DE-miRNAs were predicted using miRNet 2.0 (http://www.mirnet.ca/) (accessed on 13 February 2025) [36], which integrates data from multiple miRNA databases. Predictions were made separately for upregulated and downregulated miRNAs. The top 20 target genes for upregulated and downregulated miRNAs with high degree and betweenness were selected for further analyses.

#### 2.5.3. Protein–Protein Interaction (PPI) Network

A PPI network for DE-miRNAs’ target genes was constructed using the STRING database v12.0 (https://string-db.org/) (accessed on 13 February 2025) [37], followed by gene ontology (GO) and KEGG pathway enrichment analysis. For all comparisons, *p* < 0.05 was considered significant.

Gene ontology reference numbers were extracted from STRING database v12.0 (https://string-db.org/) from the PPI analysis and cross verified in QuickGO gene ontology and GO annotations (https://www.ebi.ac.uk/QuickGO/) (accessed on 13 February 2025).

#### 2.5.4. Network-Based Methodology for Analyzing Regulatory Interplay Among miRNA, circRNA, lncRNA, snRNA, and mRNA

An interaction network of miRNA, circRNA, lncRNA, snRNA, and mRNA was generated using miRBase v22.1 (www.mirbase.org) (accessed on 4 March 2025), separately for upregulated and downregulated miRNAs. Further, miRNA–circRNA–mRNA, miRNA–lncRNA–mRNA, and miRNA–snRNA–mRNA interaction networks were also generated.

A schematic representation of the analytical workflow is shown in Appendix A.

## 3. Results

### 3.1. Comparative Analysis of miRNA Expression in Antral Fluid from Mature Ovarian Follicles Versus Ovarian Follicular Cysts, with Sequence Similarity Insights

Semiquantitative profiling of miRNAs in antral fluid revealed 17 DE-miRNAs between OFCs and mature ovarian follicles, using a fold change threshold of ≥2 and significance (*p*-value) of ≤0.05 (Appendix A) Among these, 10 miRNAs were significantly upregulated and 7 were significantly downregulated in cysts compared to mature follicles (Figure 1).

### 3.2. Bovine and Human miRNA Sequence Similarity Insights

Nucleotide sequence similarities for the DE-miRNAs for humans and cattle are presented in Appendix A. Bovine sequences were very similar to human nucleotide sequences. Therefore, human miRNA IDs were used to construct miRNA-mRNA interaction networks and functional enrichment analyses.

### 3.3. miRNA-mRNA Interaction Analysis for Upregulated and Downregulated miRNAs

Target prediction for the 10 upregulated miRNAs identified 2171 putative target genes (Appendix A). From this list, the top 15 genes with high degree and betweenness centrality were selected for protein–protein interaction (PPI) network analysis. The PPI analysis of these genes (15 nodes and 43 edges) had significant enrichment (PPI enrichment, *p* < 0.001; Figure 2A), with 125 gene ontology (GO) biological processes and 76 significantly enriched KEGG pathways (False Discovery Rate, *p* < 0.05; Appendix A).

For the seven downregulated miRNAs, 3406 target genes were identified (Appendix A). The top 15 genes with high degree and betweenness centrality were selected for PPI network analysis. The PPI network analysis for these genes (15 nodes and 46 edges) had enrichment (PPI enrichment, *p* < 1.52 × 10^−5^; Figure 2B), with 96 GO biological processes and 49 KEGG pathways significantly enriched (False Discovery Rate, *p* < 0.05; Appendix A).

Interestingly, 1239 genes were exclusively targeted by upregulated miRNAs and 2374 genes were exclusively targeted by downregulated miRNAs, whereas 932 genes were shared targets of both groups (Figure 3). Among the shared targets, 5 genes [Ubiquitin C (*UBC*), *ELAV-Like RNA-Binding Protein 1 (ELAVL1)*, Cullin 3 (*CUL3*), *ESR1*, and MYC Proto-Oncogene (*MYC*)] were among the top 20 genes influenced by both miRNA groups (Appendix A).

### 3.4. Decoding the RNA Interactome: A Network Approach to Non-Coding and Coding RNA Interactions

Network analysis of the 10 upregulated miRNAs, including circRNA, lncRNA, snRNA, and mRNA interactions, revealed connections with 5313 unique circRNAs, 93 unique snRNAs, 237 unique lncRNAs, and 2169 unique genes (Appendix A). The interaction network is presented in Figure 4. In addition, miRNA to circRNA, miRNA to lncRNA, miRNA to snRNA, and miRNA to mRNA interactions are shown in Figure 5A, 5B, 5C, and 5D, respectively.

Network analysis of the seven downregulated miRNAs, including circRNA, lncRNAs, snRNA, and mRNA interactions, revealed connections with 5021 unique circRNAs, 39 unique snRNAs, 225 unique lncRNAs, and 3401 unique genes (Appendix A). The network is shown in Figure 6. In addition, miRNA to circRNA, miRNA to lncRNA, miRNA to snRNA, and miRNA to mRNA interactions are shown in Figure 7A, Figure 7B, Figure 7C, and Figure 7D, respectively.

The top 20 coding and non-coding RNAs with high degree and betweenness centrality targeted by up- and downregulated miRNAs are shown in Table 1.

#### 3.4.1. The Role of Circular RNAs (circRNAs) in miRNA-Mediated Gene Regulation

circRNAs act as miRNA sponges by harboring miRNA response elements (MREs), modulating gene expression through sequestration of specific miRNAs. The regulatory axis involving 10 upregulated miRNAs, 14 circRNAs, and 17 genes is shown in Figure 8A, with the corresponding gene–circRNA associations listed in Table 2. For the seven downregulated miRNAs, the circRNA–gene interaction axis included five circRNAs and five genes (Figure 8B and Table 2).

#### 3.4.2. The Role of Small Nuclear RNAs (snRNAs) in miRNA-Mediated Gene Regulation

Network analysis identified the top 10 snRNAs with high degree and betweenness centrality (Table 3). The nine miRNAs interacted with these snRNAs, forming a complex regulatory network (Figure 9). Further interactions involving snRNAs, miRNAs, and associated gene functions are shown in Figure 10.

#### 3.4.3. The Role of Long Coding RNAs (lncRNAs) in miRNA-Mediated Gene Regulation

The top 10 lncRNAs, based on degree and betweenness centrality, are shown in Table 4. Their interaction network with nine miRNAs is shown in Figure 11, and their combined interactions with miRNAs and downstream gene targets are shown in Figure 12.

## 4. Discussion

Ovarian follicular cysts in dairy cows and PCOS in women are complex reproductive disorders that, despite occurring in different species, share strikingly similar pathophysiological features. Both are characterized by anovulation, persistence of cystic ovarian follicles, hormonal imbalances (particularly androgens and insulin), and chronic low-grade inflammation. Our integrative transcriptomic analysis, encompassing miRNAs, lncRNAs, circRNAs, and associated protein-coding genes [38], uncovered a tightly interconnected gene regulatory network that governs key aspects of ovarian physiology and pathology in both conditions.

The current study was a comprehensive investigation of molecular mechanisms underlying OFCs in dairy cows, with many parallels to PCOS in women. By analyzing up- and downregulated miRNAs and their associated target genes and non-coding RNAs, we identified intricate post-transcriptional regulatory networks contributing to ovarian follicular arrest, metabolic dysfunction, and endocrine imbalance. The findings underscored the multifactorial nature of ovarian cyst formation and emphasized interplay among genetic, metabolic, and hormonal regulators that shape ovarian physiology.

### 4.1. Functional Roles of Dysregulated miRNAs in Ovarian Cyst Pathogenesis in Dairy Cows

Altered expression of miRNAs in OFCs highlighted their central role in coordinating the post-transcriptional regulation of ovarian and metabolic pathways [39]. Several dysregulated miRNAs identified in follicular cysts, including bta-miR-18a, bta-miR-30e-5p, and bta-miR-15b, are well-documented modulators of cellular stress responses (GO:0033554), insulin signaling (GO:0008286), and apoptosis (GO:0006915), processes frequently disrupted in both OFCs and PCOS [39,40,41,42,43,44,45,46,47,48,49,50,51,52,53]. For instance, miR-18a-5p promotes inflammatory cytokine signaling (GO:0019221) and cell survival in cancer biology, potentially sustaining the pathological environment that favors cyst persistence rather than regression [44,45,46]. The miR-30 family, which includes miR-30e-5p and miR-30b-5p, is deeply involved in regulating insulin receptor signaling and glucose metabolism (GO:0006006) [47,48,49]. These miRNAs likely contribute to impaired insulin sensitivity in postpartum dairy cows during NEB (GO:0052129) [50,51], a condition that mirrors metabolic dysregulation (GO:0009892) in women with PCOS. As insulin resistance (bta-04931) disrupts LH pulsatility and impairs the ovarian feedback loop, it compromises ovulatory capacity and contributes to formation of persistent, hormonally active cysts [52,53]. Holstein cows, known for their genetic predisposition to insulin resistance due to high milk production, often have altered ovarian follicular dynamics after calving, with reduced concentrations of insulin and IGF-1 [39,40,41]. This metabolic environment impairs LH responsiveness and ovulation, predisposing cows to persistent ovarian follicles and cysts.

Additionally, miR-15b-5p [42,43], miR-29a-3p [54,55], and miR-26b-5p [56] regulate follicular atresia/apoptosis (GO:0001552), extracellular matrix remodeling (GO:0085029), and steroidogenesis (GO:0006694), respectively, highlighting critical roles in ovarian structure and function [57,58,59,60,61,62,63]. These miRNAs, along with miR-191-5p [64], likely converge on key regulators of granulosa cell viability (GO:1904195) and endocrine function. Convergence of these regulatory elements on processes such as tissue remodeling (GO:0048771), angiogenesis (GO:0001525), and stress response (GO:0006950) demonstrates how subtle perturbations in miRNA networks can have broad physiological consequences [50,51,52,53].

Upregulated miRNAs in this study (e.g., bta-miR-132, bta-miR-199b, bta-miR-103, bta-miR-221, bta-miR-145) further delineated key regulatory failures in OFC pathogenesis [65,66,67,68,69,70,71]. Many of these miRNAs play roles in insulin signaling (GO:0008286), oxidative stress response (GO:0006979), and steroid hormone biosynthesis (GO:0120178) [72,73,74,75,76]. For example, miR-132-3p and miR-103a-3p target insulin signaling components, and their suppression may exacerbate the insulin-resistant state (GO:0032868) common in OFC cows [74,75]. Similarly, miR-199b-5p [67] and miR-145-5p [70,71] modulate steroidogenesis and androgen biosynthesis, with their downregulation potentially contributing to hyperandrogenism and anovulation (GO:0060280), hallmark features shared with PCOS. Moreover, downregulation of miR-193a-3p [77] and miR-29c-3p [78] may lead to unchecked inflammation (GO:0006954) [79] and oxidative damage (GO:1900408) [80], conditions impairing folliculogenesis (GO:2000355) and ovulation (GO:0060280).

The miR-26b is a potent pro-apoptotic miRNA in porcine GCs, promoting DNA damage by targeting ATM Serine/Threonine (*ATM*), a central kinase in the DNA damage response pathway, leading to increased GC apoptosis (GO:1904700) [56]. Additionally, miR-26b inhibits SMAD Family Member 4 (*SMAD4*), a key mediator of the Transforming Growth Factor Beta (TGF-β) signaling pathway (GO:0007179) that supports GC survival and function [81]. By downregulating Hyaluronan Synthase 2 (*HAS2*), miR-26b also disrupts the HA-*CD44*-caspase-3 axis, amplifying apoptotic signaling (GO:0043065) and impairing ovarian follicular development (GO:2000355) [82]. However, its downregulation in antral fluid from OFCs in the current study warrants further investigation. In contrast, upregulated bta-miR-21 functions as a protective factor, especially in mature ovarian follicles. Its upregulation following the LH surge inhibits caspase-3 activation and GC apoptosis, contributing to follicular survival and ovulation [83,84]. Transfection of mural GCs with miR-21-targeting LNA oligonucleotides increased apoptosis, further emphasizing its anti-apoptotic role in follicular maturation. Furthermore, downregulated miR-18a (a member of the miR-17-92 cluster) plays a dual role, potentially regulating apoptosis via the TGF-β/SMAD signaling pathway (GO:0007179/hsa4088), and it has also been linked to follicular atresia and altered KRAS Proto-Oncogene, GTPase (KRAS) signaling (bta-04014) in hyperstimulated conditions [85]. It represents another important player in miRNA-mediated GC fate decisions.

Recent profiling studies provided broader insights into miRNA dynamics under various physiological conditions. Both miR-103 and miR-221 were upregulated in heifers with ovarian hyperstimulation, implying roles in follicular growth (GO:0001541) or stress responses [86]. Conversely, miR-99a-3p, miR-29, and miR-145 were upregulated, implying potential involvement in follicle quiescence (GO:0044838), formation, or activation. In addition, miR-191, observed in both follicular fluid and plasma exosomes, had a linear expression pattern, indicating a stable systemic role in follicular signaling [86]. Similarly, upregulated miR-145, known for its role in ovarian follicle formation and activation, was also reduced in hyperstimulated conditions, possibly contributing to abnormal folliculogenesis [84]. Steroidogenic regulation by miRNAs is also critical. In that regard, miR-132, miR-221, and miR-199b modulated hormone production in GCs [87]. Dysregulated miR-132 influences immune (GO:0050777) and metabolic functions and is implicated in PCOS and gestational diabetes [88,89]; therefore, its increased expression in preovulatory mature follicles implied a role in late follicular phase steroidogenesis [84]. Additionally, miR-101 targets cyclooxygenase-2 (Cox2), an enzyme involved in prostaglandin synthesis (GO:0031392), inflammation, and implantation (GO:0007566); its modulation may link ovarian events to broader reproductive processes [90,91].

It should be noted that although insulin resistance is a key contributor to polycystic ovary syndrome (PCOS) in humans, and improving insulin sensitivity often alleviates PCOS symptoms, the same does not directly apply to ovarian follicular cysts (OFCs) in cattle as OFC cows are not diabetic. Whereas insulin and IGF-1 are important for normal follicular development in cows, reduced concentrations of these hormones in follicular fluids of cystic follicles are more a result of NEB and disrupted ovarian signaling than systemic insulin resistance per se. There is evidence that NEB, low insulin and IGF-1, and altered metabolic hormones contribute to anovulation and cyst formation. However, this reflects a metabolic signaling imbalance. Therefore, treatments targeting insulin resistance (e.g., insulin or metformin) used in human PCOS are not directly translatable or effective in managing OFC in cows. Strategies for OFCs focus more on nutritional management, improving energy balance (e.g., with propylene glycol in ketotic cows), and modulating local hormonal and growth factor environments.

### 4.2. Network Analysis Reveals Predicted Gene Expression Shifts in Dairy Cow Ovarian Follicular Cysts

The formation of OFCs in dairy cows is increasingly attributed to disrupted gene regulatory networks affecting hormonal signaling, cell cycle progression, apoptosis, and extracellular matrix remodeling. Our gene expression analysis identified 15 genes that were upregulated and 15 that were downregulated in OFCs, offering insights into the molecular underpinnings of these structures.

Among the upregulated miRNAs target genes, several have roles in promoting cell survival and proliferation. For instance, MYC Proto-Oncogene (*MYC*) and Cyclin-Dependent Kinase 6 (*CDK6*) are involved in driving granulosa cell proliferation and activating ovulatory pathways, but their overexpression in the absence of proper differentiation may result in cyst formation [92,93]. Similarly, Phosphatase and Tensin Homolog (*PTEN*) and Epidermal Growth Factor Receptor (*EGFR*), regulatory subunits in the PI3K/AKT pathway (bta-04151), are associated with enhanced insulin and growth factor signaling, which is often dysregulated in cystic ovarian conditions [94]. Overexpression of *ESR1*, encoding the estrogen receptor alpha, may impair feedback mechanisms that regulate LH surges, contributing to anovulation [95]. Upregulated Regulatory Factor (*RFX*) 3 and *RFX4* may further disrupt neuroendocrine signaling, affecting the timing and amplitude of the LH peak necessary for antral follicular rupture [96]. Additional upregulated genes such as *UBC*, *CUL3*, and Heat Shock Protein 90 Alpha Family Class A Member 1 (*HSP90AA1*) are involved in protein homeostasis. *UBC* and *CUL3* contribute to the ubiquitin–proteasome system, regulating the degradation proteins related to the cell cycle and steroidogenesis, whereas *HSP90AA1* is a molecular chaperone stabilizing key steroid hormone receptors. Dysregulation of these genes may lead to prolonged receptor signaling or degradation errors, exacerbating follicular persistence [97]. *FN*1 overexpression suggests excessive extracellular matrix remodeling, potentially impeding follicle rupture, whereas Growth Factor Receptor-Bound Protein 2 (*GRB2*), *ELAVL1*, *Trinucleotide Repeat-Containing Adaptor 6B (TNRC6B)*, and Ataxin 1 (*ATXN1*) influence mRNA stability, post-transcriptional regulation, and signal transduction, likely reinforcing pathological gene expression profiles in cystic follicles.

Conversely, several downregulated miRNAs target genes that indicate a failure in apoptotic pathways and growth regulation. Notably, *PTEN*, BCL2 Apoptosis Regulator *(BCL)2*, MCL1 Apoptosis Regulator, BCL2 Family Member (*MCL1*), and Thrombospondin 1 (*THBS1*) are all key regulators of cell survival and programmed cell death. Their reduced expression suggests impaired atresia, allowing nonovulatory follicles to persist [98]. Downregulation of *CDK6* and Cyclin-Dependent Kinase Inhibitor 1A (*CDKN1A*) disrupts cell cycle checkpoints, which could destabilize the balance between proliferation and differentiation in granulosa cells. Interestingly, genes like *UBC*, *CUL3*, *ELAVL1*, *ESR1*, and *MYC* appeared in both upregulated and downregulated gene sets, likely reflecting tissue-specific expression or complex feedback mechanisms within various follicular compartments. Furthermore, Trinucleotide Repeat-Containing Adaptor 6A (*TNRC6A*) and SERPINE1 MRNA-Binding Protein 1 (*SERBP1*), involved in post-transcriptional regulation via RNA-binding and miRNA pathways, may contribute to altered gene silencing dynamics. Decreased expression of Protein Tyrosine Phosphatase 4A1 (*PTP4A1*) and Tubulin Beta Class I (*TUBB*), associated with cell migration and cytoskeletal integrity, suggests impaired follicular remodeling and ovulatory capacity. Collectively, these changes reflected a molecular shift favoring unchecked growth, prolonged survival, and failed ovulation, hallmarks of OFC pathology.

Concurrent dysregulation of growth-promoting and apoptotic genes emphasized the multifactorial nature of OFC development and underscored the need for targeted molecular interventions to restore normal ovarian follicular dynamics in affected dairy cows.

### 4.3. Integrated Analysis of miRNA-Regulated Coding and Non-Coding RNAs in Ovarian Follicular Cysts

Target analysis of these miRNAs identified a suite of coding and non-coding RNAs [e.g., Nuclear Enriched Abundant Transcript 1 (*NEAT1*), Potassium Voltage-Gated Channel Subfamily Q Member 1 Opposite Strand/Antisense Transcript 1 (*KCNQ1OT1*), Taurine-Upregulated 1 (*TUG1*), and X Inactive Specific Transcript (*XIST*)] that function in gene silencing, chromatin remodeling, cell proliferation, and metabolic homeostasis [55,99,100,101,102,103,104,105,106]. Notably, several genes such as *PTEN* [104,105], *CDK6* [93], and Vascular Endothelial Growth Factor A (*VEGFA*) [107,108] are well-characterized modulators of ovarian function. *PTEN* regulates the PI3K-AKT signaling axis, a central pathway in ovarian follicular growth and hormone signaling [104,105], whereas *VEGFA* mediates ovarian angiogenesis [107,108]. Their dysregulation likely impairs ovarian follicular development and contributes to cyst formation [109]. The presence of Argonaute RISC Catalytic Component 2 (*AGO2*) [93,101,102,103] and Dicer 1 (*DICER1*) [110,111,112], core components of miRNA biogenesis, among the top targets reinforced the possibility of a feedback disruption within the miRNA regulatory system, creating a self-perpetuating loop of dysregulation [103,104,105].

The target coding and non-coding RNAs of downregulated miRNAs, including *CDK6*, *NEAT1*, *XIST*, Cyclin D1 (*CCND1*), and ATPase Sarcoplasmic/Endoplasmic Reticulum Ca^2+^ Transporting 2 (*ATP2A2*), highlighted further disruptions in cell cycle control, calcium homeostasis, and transcriptional regulation [93,98,113,114,115,116]. Of particular interest was the dual targeting of *NEAT1* [98], *XIST* [113,114], *KCNQ1OT1* [99,100,115,116], and *CDK6* [93] by both up- and downregulated miRNAs, suggesting a central regulatory hub where multiple signaling pathways intersect to govern ovarian fate [117,118,119]. Such convergence may make these molecules promising therapeutic targets or biomarkers for the early diagnosis of OFCs and PCOS.

### 4.4. Emerging Role of Circular RNAs in Post-Transcriptional Control of Bovine Ovarian Cyst Pathways

Our findings also underscored the emerging role of circular RNAs (circRNAs) as pivotal regulators of gene expression in ovarian physiology. Incorporation of circular RNAs (circRNAs) into this study provided novel insights into ceRNA (competing endogenous RNA; [38,120]) networks. CircRNAs such as hsa_circ_0007694 and hsa_circ_0007945 interact with miRNAs regulating critical pathways like PI3K-AKT, insulin signaling, and steroid biosynthesis, reinforcing their potential role as post-transcriptional buffers that fine-tune ovarian gene expression [121,122,123]. Their sponge-like behavior may amplify or suppress specific miRNA effects, indirectly impacting granulosa cell apoptosis, ovarian follicular development, and androgen production. The association of *ATM* [124] and ALF Transcription Elongation Factor 4 (*AFF4*) with these circRNAs suggests a link between DNA repair, oxidative stress, and transcriptional elongation processes in follicular health [125,126].

In the current study, network analysis revealed that the interactions of miR-103 with the ADAM10 circRNA-gene, miR-132 with the SIRT1 circRNA-gene, and miR-193-3p with the IGFBP5 circRNA-gene were associated with increased proliferation and inflammation. Similarly, interactions of miR-145 with the STAT3 circRNA-gene, as well as miR-29c, miR-221, and miR-103-3p with DICER1, were linked to inflammation and apoptosis.

### 4.5. Integrated ncRNA and Proteostasis Network Disruption in Bovine Ovarian Cysts

Expanding beyond canonical ncRNAs, our study also identified pseudogenes such as *RN7SKP* and *RNA5SP*, and other small non-coding RNAs, (Small Nucleolar RNA, C/D Box (*SNORD*) 17, *SNORD69*), highlighting the epigenetic complexity of cystic ovarian disease. RN7SK Pseudogene 91 (*RN7SKP91*) is likely a player in the FSH signaling pathway, which is essential for ovarian follicular development [127,128,129], perhaps by promoting follicle growth, oocyte maturation, and estrogen production [130,131]. These elements may subtly modulate transcriptional dynamics or ribosomal function, thereby contributing to altered gene expression profiles present in OFCs and PCOS [127,131]. Although precise mechanisms remain to be fully characterized, their consistent presence in dysregulated ovarian transcriptomes warrants further investigation. Furthermore, based on the dysregulated expression of genes involved in protein degradation, such as Ubiquitin Protein Ligase E3C (*UBE3C*), OTU Deubiquitinase 4 (*OTUD4*), and Proteasome 26S Subunit, Non-ATPase 7 (*PSMD7*), impaired proteostasis could be a contributing factor to granulosa cell dysfunction and follicular cyst formation. Accumulation of damaged or misfolded proteins may interfere with cellular homeostasis and amplify metabolic and oxidative stress responses, further compromising follicular health [132,133,134,135,136]. The study also emphasized contributions of transcriptional and metabolic regulators such as Glycogen Synthase Kinase 3 Beta (*GSK3B*), TSC Complex Subunit 1 *(TSC1*), and Zinc Finger Protein (*ZNF*) family members (e.g., *ZNF121*, *ZNF460*) in orchestrating granulosa cell proliferation, apoptosis, and hormonal feedback [137,138,139,140,141,142,143]. Based on these findings, we inferred that OFCs arise not from a single regulatory failure but rather from a cascade of interconnected disturbances affecting transcription, translation, signaling, and cellular metabolism.

### 4.6. Shared Molecular Pathways Link Ovarian Follicular Cysts in Dairy Cows and PCOS in Women

Parallels between OFCs and PCOS were evident at both the molecular and systemic levels. Both conditions involve insulin resistance, hyperandrogenism, altered ovarian folliculogenesis, and chronic low-grade inflammation. In dairy cows, NEB during early lactation intensifies these effects, reducing insulin and IGF-1 concentrations and impairing LH-mediated ovulation [144,145,146]. Hyperandrogenism, coupled with impaired apoptosis and dysregulated cell cycle genes (e.g., *CCND1*, *CDK6*), prevents proper ovarian follicular selection and dominance, leading to cyst formation [93,147,148,149]. This mirrors PCOS pathophysiology in women, where chronic anovulation and ovarian hyperthecosis contribute to similar reproductive dysfunction. Clear associations between reproductive and metabolic health emphasize the need for integrative approaches that consider both endocrine and metabolic contexts. Regulatory networks uncovered in this study exemplify how local transcriptomic changes are both a reflection of and a response to systemic physiological stress.

### 4.7. Translational Insights into PCOS from miRNA Profiling of Ovarian Cysts in Cows

Taken together, molecular and functional similarities between OFCs and PCOS establish dairy cows as a compelling model for studying human ovarian disorders. The high metabolic demands of lactation and the resulting insulin-resistant state in postpartum cows replicate many features of PCOS, including disrupted folliculogenesis, hyperandrogenism, and ovulatory failure [41,150,151]. By leveraging this model, researchers could investigate conserved molecular mechanisms in a naturally occurring disease system, offering insights into biomarker development and novel therapeutic targets. Moreover, improving reproductive efficiency in dairy cattle holds substantial economic value, creating a unique synergy between animal and human health research.

It is important to highlight that, unlike human polycystic ovary syndrome (PCOS), where hyperandrogenism is a systemic hallmark, ovarian follicular cysts (OFCs) in cattle are primarily associated with localized, intrafollicular androgen excess. A naturally occurring form of this condition, characterized by elevated levels of androstenedione, has been documented and is referred to as the High A4 phenotype in cows [152]. These cows have upregulated mRNA expression of *LHCGR*, *CYP11A1*, and *CYP17A1* in theca cells, a gene expression pattern that parallels some molecular features observed in women with PCOS [152,153]. Consistent findings in human studies include increased expression of *CYP11A1*, *CYP17A1*, GATA-Binding Protein 6 (*GATA6*), and *LHCGR* in theca cells from PCOS patients [154], mirroring the gene profile seen in High A4 cows [152]. In granulosa cells from High A4 cows, microRNA profiles indicate the suppression of genes involved in cell cycle regulation, along with signs of cell cycle arrest when compared to controls [155]. Complementary in vitro studies have shown that A4 exposure inhibits granulosa cell proliferation while enhancing the secretion of anti-Müllerian hormone (AMH) and the expression of Catenin Beta-Interacting Protein 1 (*CTNNBIP1*) mRNA—factors potentially involved in follicular arrest [155]. Furthermore, follicles from High A4 cows failed to progress in culture over a seven-day period, unlike those from control cows [152,153]. The ovarian cortex of High A4 cows also displayed increased fibrotic tissue and oxidative stress markers, accompanied by higher secretion of A4 and other steroid hormones relative to control animals [153]. Interestingly, fibrotic thickening of the ovarian stroma, commonly resulting from collagen accumulation, is a recognized histological feature of PCOS in women [156]. This similarity supports the idea that increased ovarian fibrosis is a shared pathological trait between High A4 cows and women with PCOS, and it may also be present in cows with OFCs.

Key miRNAs, categorized by their functions in OFCs and PCOS, are shown in Table 5 and Table 6, respectively. Further, comparative aspects of miRNAs’ roles in bovine follicular cysts versus PCOS in women are presented in Table 7.

#### 4.7.1. MicroRNA-Based Parallels Between Bovine Follicular Cysts and Human PCOS: A Translational Perspective

Ovarian follicular cysts in dairy cows and PCOS in women are both important reproductive disorders, though they manifest differently between species. A comparative examination revealed shared and divergent pathophysiological mechanisms, especially concerning disrupted ovarian folliculogenesis, anovulation, and hormonal imbalances [5,23,157]. One of the most promising areas of current research involves the role of microRNAs (miRNAs) in regulating these reproductive functions and the potential to use these molecules as therapeutic targets or diagnostic biomarkers.

##### Comparative Pathophysiology: Similarities and Differences

At the core of both OFCs in cows and PCOS in women is disrupted ovarian follicular development leading to anovulation. In dairy cows, OFCs arise primarily due to altered secretion patterns of LH and follicle-stimulating hormone (FSH), leading to insufficient LH surges and persistence of anovulatory follicles [5,8]. Similarly, women with PCOS often have chronic anovulation due to an elevated LH-to-FSH ratio, often accompanied by increased androgen concentrations [158,159,160]. Despite the commonality of follicular arrest, PCOS is characterized by pronounced hyperandrogenism and insulin resistance, features less prominent or rare in cattle with OFCs.

The role of metabolic stress also varies. In dairy cows, NEB, particularly during the postpartum period, contributes to cyst development, whereas in women, lifestyle factors such as obesity and chronic stress influence the onset and severity of PCOS [161,162]. Moreover, OFCs in cows can often be resolved with short-term hormonal therapy or nutritional adjustments; in contrast, PCOS typically requires long-term, multifaceted management strategies, including pharmacological interventions and lifestyle modification [8,162,163,164].

##### The Role of microRNAs in Ovarian Dysfunction

Various miRNAs have emerged as critical regulators of ovarian function. Both bovine and human studies have identified miRNA-regulated pathways, including TGF-β and PI3K/Akt signaling, that are integral to ovarian folliculogenesis, steroidogenesis, and granulosa cell viability [42,45,53,165]. Aberrant miRNA expression disrupts these pathways, contributing to OFC formation and ovarian dysfunction.

In dairy cows, targeting specific miRNAs offers a novel strategy for managing follicular cysts. For example, as overexpression of miR-21 is associated with the survival of granulosa cells in follicular cysts [166], inhibiting this miRNA could promote atresia of persistent follicles and restore normal ovulatory cycles. Similarly, supplementation with miR-26a mimics, which modulate the SMAD/TGF-β pathway, may enhance ovarian follicular development and ovulation rates [167,168]. These interventions could complement existing approaches, such as improving nutritional status to resolve NEB, ultimately reducing the prevalence of COD.

In women with PCOS, miRNAs also regulate key features such as insulin resistance and hyperandrogenism. Inhibiting miR-193, for instance, has been proposed to improve insulin sensitivity, potentially ameliorating a primary metabolic dysfunction associated with PCOS [169]. Conversely, restoring expression of miR-132 may overcome follicular arrest and promote ovulation [170,171]. Advanced approaches, including exosomal miRNAs, are being explored to modify the ovarian microenvironment and restore follicular development.

##### Challenges in miRNA-Based Therapeutics

Despite their potential, miRNA-based therapies face hurdles [172,173,174]. For example, delivering miRNAs specifically to ovarian tissue without affecting other organs is a primary challenge. Current research is focused on developing targeted delivery systems, e.g., nanoparticles or engineered exosomes, to ensure efficacy while minimizing off-target effects. Furthermore, the long-term safety of such therapies is not well understood, necessitating comprehensive preclinical and clinical studies to assess therapeutic benefit and potential risks.

*miRNAs as diagnostic biomarkers*: miRNAs also have potential as biomarkers for diagnosis and monitoring of ovarian disorders. In dairy cows, circulating concentrations of miRNAs (e.g., miR-21, miR-26a, and miR-130b) have been associated with cystic ovarian follicles [53], making them viable candidates for early detection. These could be used in conjunction with ultrasonography to improve diagnostic accuracy and allow timely intervention. Similarly, in women with PCOS, specific miRNAs correlate with disease severity and metabolic symptoms. For instance, elevated miR-93 may indicate insulin resistance, whereas variations in miR-132 can reflect follicular activity and ovulatory potential. Longitudinal monitoring of these miRNAs could provide insights into treatment responses, enabling therapy tailored to individual patient profiles. Moreover, miRNA profiling may classify PCOS into subtypes, enabling personalized management.

##### Cross-Species Comparative Research and Its Benefits

A key advantage of studying both species lies in the opportunity for translational research. Shared molecular pathways, particularly those involving miRNAs, offer insights that can be applied across species [175]. Research in dairy cows can illuminate aspects of steroidogenesis and follicular maturation that are relevant to human medicine [176]. Conversely, human research into insulin signaling and androgen regulation may inform bovine reproductive strategies [177].

Collaborative research efforts that bridge veterinary and human medical science could accelerate the development of miRNA-based diagnostics and therapeutics [177]. Identifying conserved mechanisms of ovarian dysfunction informs interventions that address fundamental causes rather than symptoms or clinical signs, thereby improving reproductive outcomes in women and animals [178].

## 5. Conclusions

This study provided an integrative, cross-species analysis of OFCs in dairy cows, identifying molecular parallels with PCOS in women. Through transcriptomic profiling and systems-level network analysis, we identified a core set of miRNAs—including miR-18a-5p, miR-30e-5p, miR-26b, and miR-21—and key regulatory genes such as *MYC*, *ESR1*, *PIK3R1*, *PTEN*, and *CDK6* that govern essential processes, including granulosa cell function, steroidogenesis, insulin signaling, and extracellular matrix remodeling. Dysregulation of non-coding RNAs, including lncRNAs (e.g., *NEAT1*, *KCNQ1OT1*), circRNAs, and pseudogenes, further underscores the multifactorial nature of these ovarian disorders.

Importantly, both OFCs and PCOS are marked by disruptions in conserved pathways such as TGF-β/SMAD, PI3K-Akt, and oxidative stress signaling, highlighting shared pathophysiological mechanisms across species. Convergence of transcriptional and post-transcriptional regulatory disruptions, particularly involving miRNAs, highlights their central role in deranged ovarian homeostasis and folliculogenesis. These findings not only enhance our understanding of OFCs in livestock but also offer valuable insights into human reproductive disorders.

The translational potential of miRNA- and circRNA-based diagnostics and therapeutics is particularly promising, although challenges remain in targeted delivery and long-term safety. Future research should focus on validating these molecular targets in vivo, exploring environmental and nutritional modulators and developing precise RNA delivery platforms. Ultimately, by bridging animal and human reproductive biology, this study lays a foundation for innovative, species-specific interventions that could transform ovarian disorder management and improve reproductive health outcomes across species.

## Figures and Tables

**Figure 1 genes-16-00791-f001:**
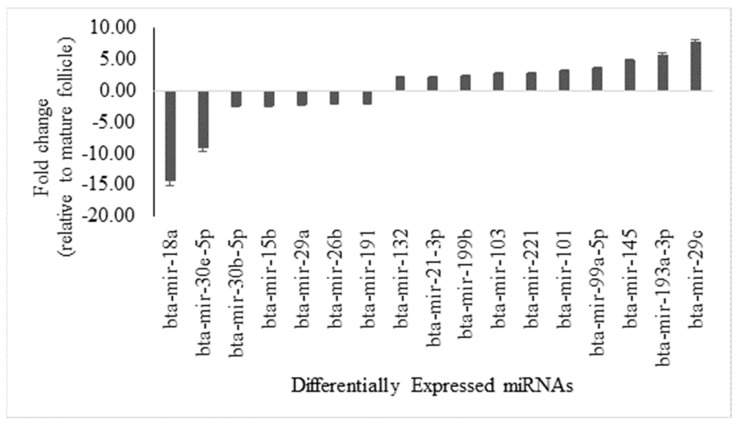
Fold regulation of differentially expressed miRNAs in antral fluid collected from ovarian follicular cysts versus mature ovarian follicles in Holstein dairy cows. Of the 84 bovine-specific well-characterized miRNAs that were investigated, 10 were greater (*p* ≤ 0.05; fold ≥2) and 7 were lower (*p* ≤ 0.05; fold ≤−2) in follicular cysts.

**Figure 2 genes-16-00791-f002:**
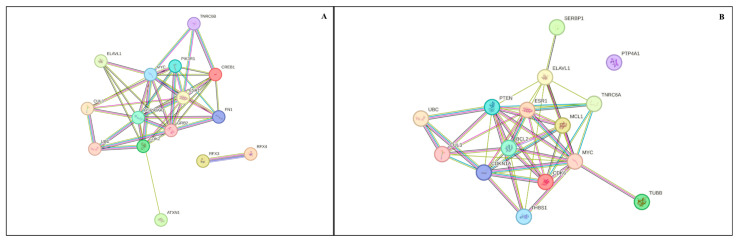
Differentially expressed miRNAs in antral fluid collected from ovarian follicular cysts versus mature ovarian follicles in Holstein dairy cows. (**A**) STRING protein–protein interaction (PPI) network. Upregulated miRNAs predicted the top 15 target genes; their interaction network had 15 nodes and 43 edges (PPI enrichment, *p* = 0.001). (**B**) STRING protein–protein interaction (PPI) network. Downregulated miRNAs predicted the top 15 target genes, and their interaction network had 15 nodes and 46 edges (PPI enrichment, *p* < 1.52 × 10^−5^). Color nodes represent proteins, whereas edges (lines) represent interactions.

**Figure 3 genes-16-00791-f003:**
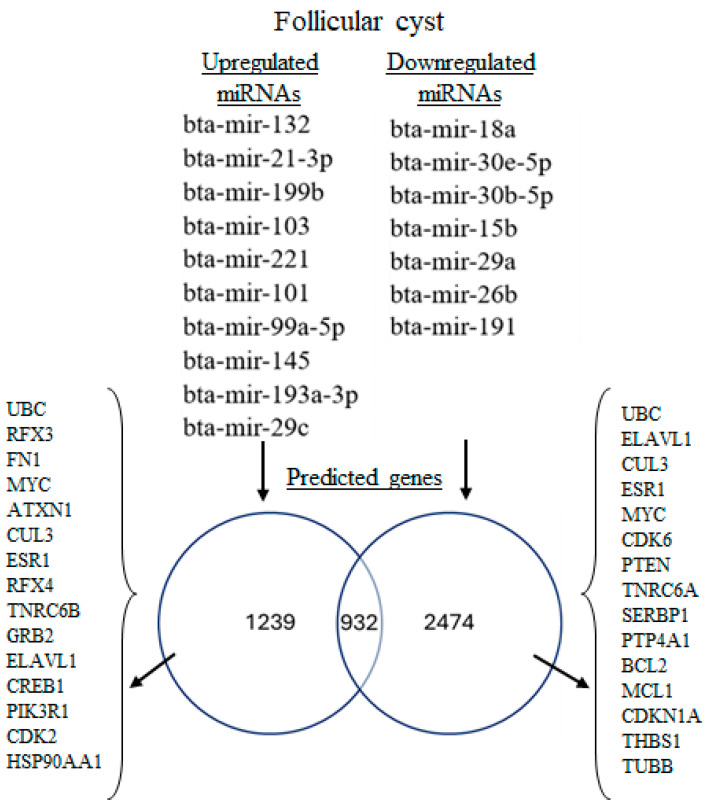
Differentially expressed miRNAs in antral fluid collected from ovarian follicular cysts versus mature ovarian follicles in Holstein dairy cows. Upregulated miRNAs targeted 1239 genes and downregulated miRNAs targeted 2474 genes, whereas 932 genes were targeted by both upregulated and downregulated miRNAs. The top 15 genes regulated by up- and downregulated miRNAs were identified through network centrality analysis (Appendix A). *UBC*, Ubiquitin C; *RFX3*, Regulatory Factor X3; *FN1*, Fibronectin 1; *MYC*, MYC Proto-Oncogene; BHLH Transcription Factor; ATXN1, Ataxin 1; *CUL3*, Cullin 3; *ESR1*, Estrogen Receptor 1; *RFX4*, Regulatory Factor X4; *TNRC6B*, Trinucleotide Repeat-Containing Adaptor 6B; *GRB2*, Growth Factor Receptor-Bound Protein 2; *ELAVL1*, ELAV-Like RNA-Binding Protein 1; *CREB1*, CAMP-Responsive Element-Binding Protein 1; *PIK3R1*, Phosphoinositide-3-Kinase Regulatory Subunit 1; *CDK2*, Cyclin-Dependent Kinase 2; *HSP90AA1*, Heat Shock Protein 90 Alpha Family Class A Member 1; *CDK6*, Cyclin-Dependent Kinase 6; *PTEN*, Phosphatase and Tensin Homolog; *TNRC6A*, Trinucleotide Repeat-Containing Adaptor 6A; *SERBP1*, SERPINE1 MRNA-Binding Protein 1; *PTP4A1*, Protein Tyrosine Phosphatase 4A1; *BCL2*, BCL2 Apoptosis Regulator; *MCL1*, MCL1 Apoptosis Regulator, BCL2 Family Member; CDKN1A, Cyclin-Dependent Kinase Inhibitor 1A; *THBS1*, Thrombospondin 1; *TUBB*, Tubulin Beta Class I.

**Figure 4 genes-16-00791-f004:**
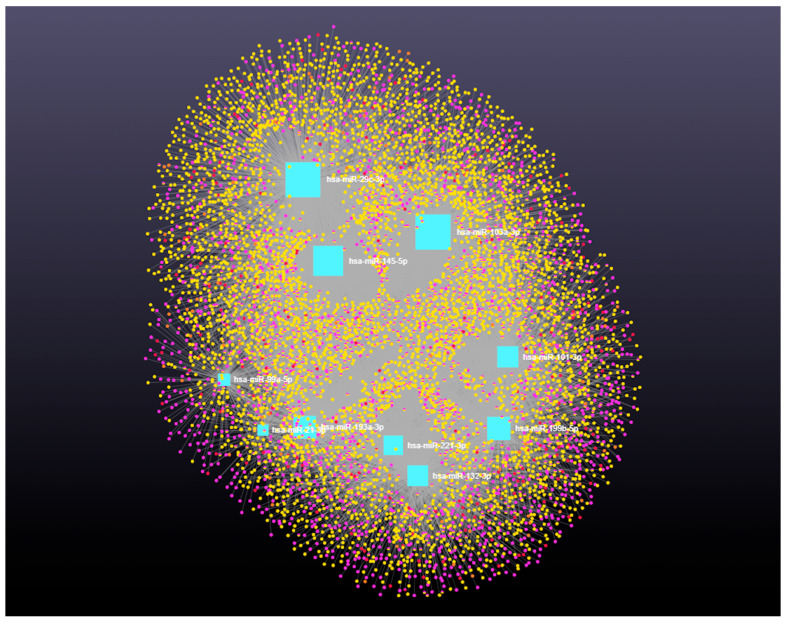
Differentially expressed miRNAs in antral fluid collected from ovarian follicular cysts versus mature ovarian follicles in Holstein dairy cows. Network of interactions among upregulated miRNAs in follicular cyst fluid and their interacting circRNAs, lncRNAs, snRNAs, and mRNAs. Blue squares represent upregulated miRNAs, purple circles represent circRNAs, orange circles represent lncRNAs, red circles represent snRNAs, and yellow circles represent genes.

**Figure 5 genes-16-00791-f005:**
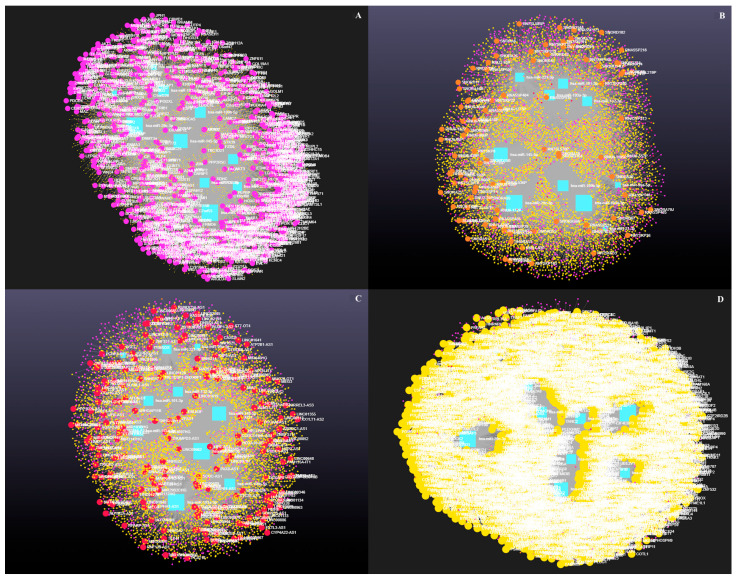
Differentially expressed miRNAs in antral fluid collected from ovarian follicular cysts versus mature ovarian follicles in Holstein dairy cows. (**A**). Interaction network of upregulated miRNAs and circRNAs. Blue squares represent miRNAs and purple circles represent circRNAs. (**B**). Interaction network of upregulated miRNAs and lncRNAs. Blue squares represent miRNAs, and orange circles represent lncRNAs. (**C**). Interaction network of upregulated miRNAs and snRNAs. Blue squares represent miRNAs and red circles represent snRNAs. (**D**). Interaction network of upregulated miRNAs and mRNAs. Blue squares represent miRNAs and yellow circles represent genes. A complete list of interactions involving upregulated miRNAs, circRNAs, lncRNAs, snRNAs, and mRNAs is provided in Appendix A.

**Figure 6 genes-16-00791-f006:**
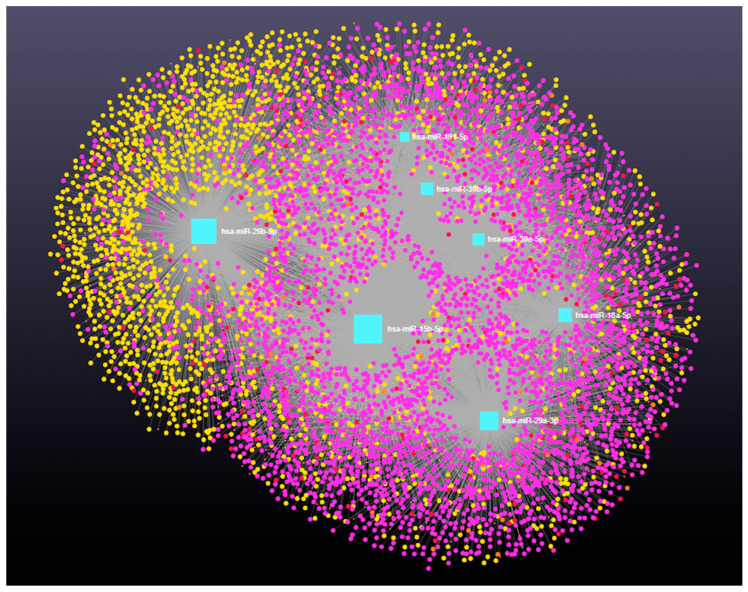
Differentially expressed miRNAs in antral fluid collected from ovarian follicular cysts versus mature ovarian follicles in Holstein dairy cows. Network of interactions among downregulated miRNAs and their interacting circRNAs, lncRNAs, snRNAs, and mRNAs. Blue squares represent downregulated miRNAs, purple circles represent circRNAs, orange circles represent lncRNAs, red circles represent snRNAs, and yellow circles represent genes.

**Figure 7 genes-16-00791-f007:**
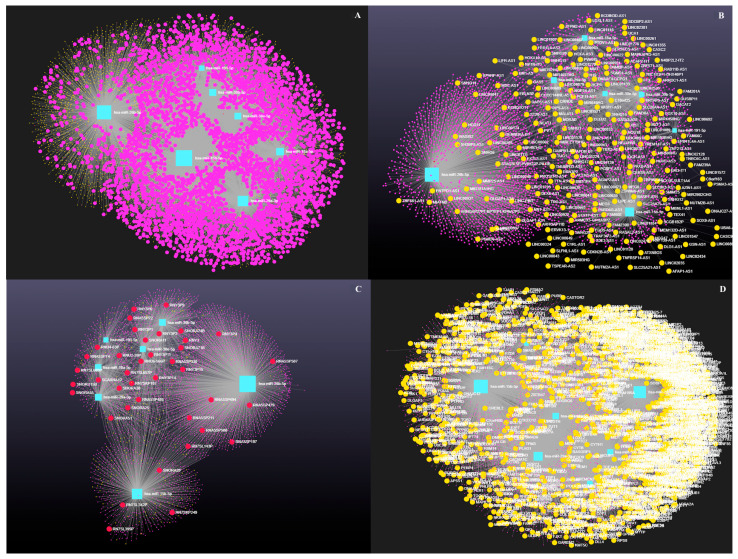
Differentially expressed miRNAs in antral fluid collected from ovarian follicular cysts versus mature ovarian follicles in Holstein dairy cows. (**A**)**.** Interaction network of downregulated miRNAs and circRNAs. Blue squares represent miRNAs and purple circles represent circRNAs. (**B**). Interaction network of downregulated miRNAs and lncRNAs. Blue squares represent miRNAs and orange circles represent lncRNAs. (**C**). Interaction network of downregulated miRNAs and snRNAs. Blue squares represent miRNAs and red circles represent snRNAs. (**D**). Interaction network of downregulated miRNAs and mRNAs. Blue squares represent miRNAs, and yellow circles represent genes. A complete list of interactions involving downregulated miRNAs, circRNAs, lncRNAs, snRNAs, and mRNAs is provided in Appendix A.

**Figure 8 genes-16-00791-f008:**
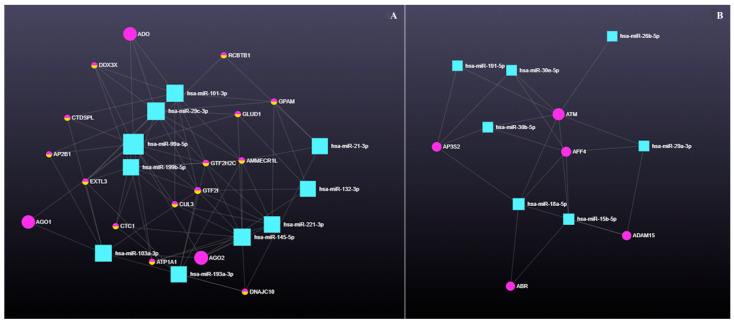
(**A**) Up- and (**B**) downregulated miRNA and circ–gene axis interactions. Differentially expressed miRNAs in antral fluid collected from ovarian follicular cysts versus mature ovarian follicles in Holstein dairy cows. Blue squares represent miRNAs, and purple and purple/yellow circles represent circ–genes.

**Figure 9 genes-16-00791-f009:**
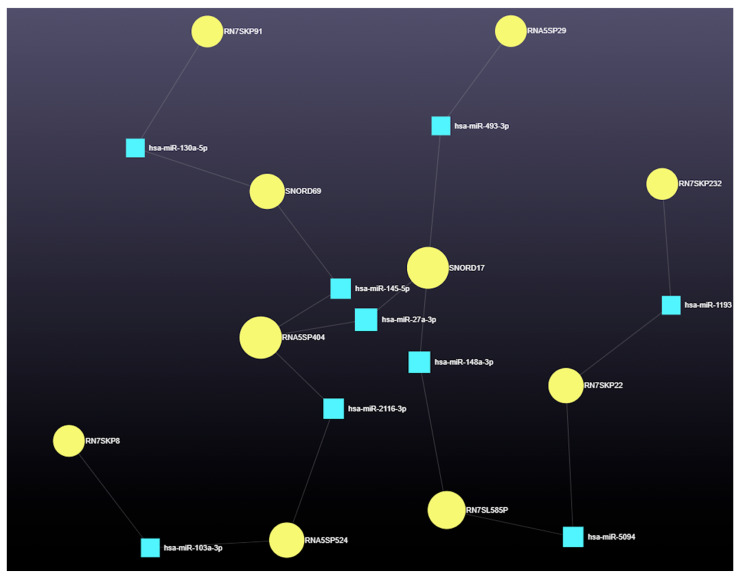
Network interaction of 9 miRNAs with snRNAs. Differentially expressed snRNAs in antral fluid collected from ovarian follicular cysts versus mature ovarian follicles in Holstein dairy cows. Blue squares represent miRNAs, and yellow circles represent snRNAs.

**Figure 10 genes-16-00791-f010:**
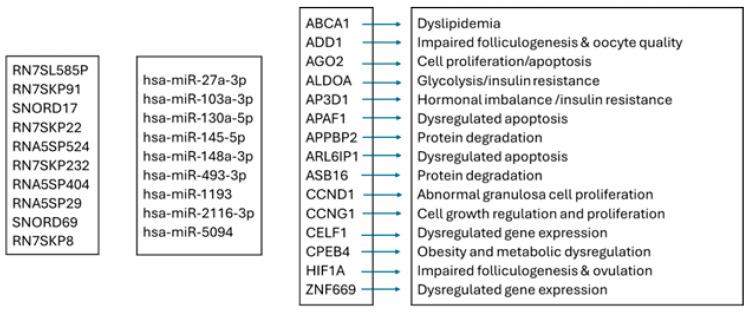
Flow chart: interactions among snRNAs, miRNAs, genes, and their functions. RN7SL585P—RNA, 7SL, Cytoplasmic 585, Pseudogene; RN7SKP91—RN7SK Pseudogene 91; SNORD17—Small Nucleolar RNA, C/D Box 17; RN7SKP22—RN7SK Pseudogene 22; RNA5SP524—RNA, 5S Ribosomal Pseudogene 524; RN7SKP232—RN7SK Pseudogene 232; RNA5SP404—RNA, 5S Ribosomal Pseudogene 404; RNA5SP29—RNA, 5S Ribosomal Pseudogene 29; SNORD69—Small Nucleolar RNA, C/D Box 69; RN7SKP8—RN7SK Pseudogene 8; *ABCA1*, ATP-Binding Cassette Subfamily A Member 1; *ADD1*, Adducin 1; *AGO2*, Argonaute RISC Catalytic Component 2; *ALDOA*, Aldolase, Fructose-Bisphosphate A; *AP3D1*, Adaptor-Related Protein Complex 3 Subunit Delta 1; *APAF1*, Apoptotic Peptidase Activating Factor 1; *APPBP2*, Amyloid Beta Precursor Protein-Binding Protein 2; *ARL6IP1*, ARL6-Interacting Reticulophagy Regulator 1; *ASB16*, Ankyrin Repeat- and SOCS Box-Containing 16; *CCND1*, Cyclin D1; *CCNG1*, Cyclin G1; *CELF1*, CUGBP ELAV-Like Family Member 1; *CPEB4*, Cytoplasmic Polyadenylation Element-Binding Protein 4; *HIF1A*, Hypoxia-Inducing Factor 1 A; *ZNF669*, Zinc Finger Protein 669.

**Figure 11 genes-16-00791-f011:**
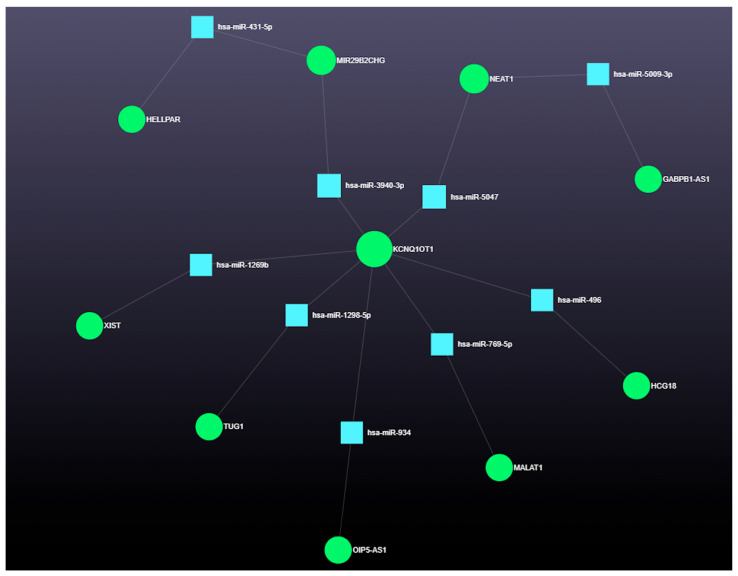
Network interaction of lncRNAs with 9 miRNAs. Blue squares represent miRNAs, and green circles represent lncRNAs.

**Figure 12 genes-16-00791-f012:**
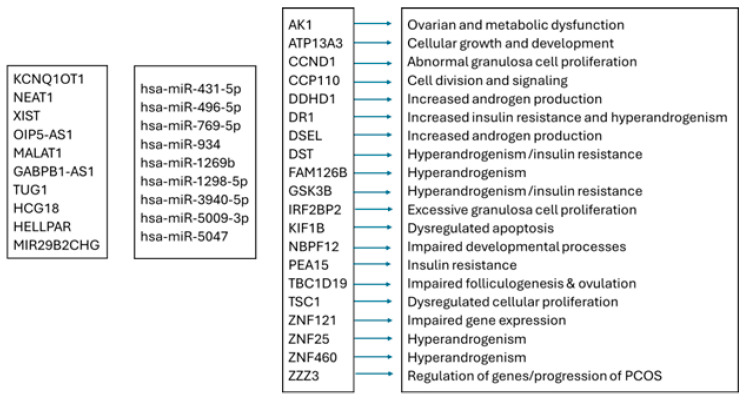
Flow chart: interactions among lncRNAs, miRNAs, genes, and their functions. *KCQN1OT1*, KCNQ1 Opposite-Strand/Antisense Transcript 1; *XIST*, X Inactive Specific Transcript; *OIP5-AS1*, Opa-Interacting Protein 5 Antisense RNA 1; *HCG18*, Major Histocompatibility Complex Group 18; *MALAT1*, Metastasis-Associated Lung Adenocarcinoma Transcript 1; *GABPB1-AS1*, GA-Binding Protein Transcription Factor Subunit Beta 1, Antisense RNA 1; *HELLPAR*, HELLP-Associated Long Non-Coding RNA; *TUG1*, Taurine-Upregulated 1; *MIR29B2CHG*, MIR29B2 and MIR29C Host Gene; *AK1*, Adenylate Kinase 1; *ATP13A*3, ATPase 13A3; *CCND1*, Cyclin D1; *CCP110*, Centriolar Coiled-Coil Protein 110; *DDHD1*, DDHD Domain-Containing 1; *DR1*, Downregulator of Transcription 1; *DSEL*, Dermatan Sulfate Epimerase-Like; *DST*, Dystonin; *FAM126B*, Family with sequence similarity 126 member B; *GSK3B*, Glycogen Synthase Kinase 3 Beta; *IRF2BP2*, Interferon Regulatory Factor 2 Binding Protein 2; *KIF1B*, Kinesin Family Member 1B; *NBPF12*, NBPF Member 12; *PEA15*, Proliferation and Apoptosis Adaptor Protein 15; *TBC1D19*, TBC1 Domain Family Member 19, *TSC1*, TSC Complex Subunit 1; *ZNF121*, Zinc Finger Protein 121; *ZNF460*, Zinc Finger Protein 460; *ZZZ3*, Zinc Finger ZZ-Type Containing 3.

**Table 1 genes-16-00791-t001:** Top up- and downregulated miRNAs-predicted genes and non-coding RNAs in ovarian follicular cysts, identified through network centrality analysis.

Gene ID	Degree	Betweenness
**Upregulated**		
** *NEAT1* **	7	83,015.76
** *KCNQ1OT1* **	6	59,933.59
*AGO2*	5	61,059.11
*NUFIP2*	5	41,540.83
*PTEN*	5	34,929.53
*TUG1*	5	38,981.89
** *XIST* **	5	43,828.79
*HCG18*	5	41,237.25
*DICER1*	4	22,365.89
*E2F3*	4	21,041.56
*FZD6*	4	27,243.63
*HIF1A*	4	21,041.56
*MYCN*	4	23,217
*REL*	4	23,217
*SNHG1*	4	28,911.3
*VEGFA*	4	20,431.3
** *CDK6* **	4	22,842.08
*SIRT1*	4	26,887.84
*EGFR*	4	22,653.18
*PSMA3-AS1*	4	24,494.06
**Downregulated**		
** *KCNQ1OT1* **	6	62,381.99
** *NEAT1* **	6	62,381.99
*OIP5-AS1*	5	40,151.9
*RNF138*	5	22,458.95
*CCND1*	4	38,215.53
*CCND2*	4	37,271.94
** *CDK6* **	4	37,271.94
*HNRNPM*	4	37,271.94
*SERBP1*	4	33,966.8
** *XIST* **	6	33,717.75
*EEF1A1*	4	31,595
*ATP2A2*	4	26,749.51
*CREG1*	4	26,749.51
*KLHDC10*	4	26,749.51
*MLXIP*	4	26,749.51
*PPTC7*	4	26,749.51
*PSMD7*	4	26,749.51
*SRPR*	4	26,749.51
*SRPRA*	4	26,749.51

Coding and non-coding RNAs in bold letters are same and indicate that they were co-targeted by both up- and downregulated miRNAs. A complete list of up- and down-regulated miRNAs, circRNAs, lncRNAs, snRNAs, and mRNAs interactions is provided in Appendix A. *NEAT1*, Nuclear Paraspeckle Assembly Transcript 1; *KCQN1OT1*, KCNQ1 Opposite-Strand/Antisense Transcript 1; *AGO2*, Argonaute RISC Catalytic Component 2; *NUFIP2*, Nuclear FMR1-Interacting Protein 2; *PTEN*, Phosphatase and Tensin Homolog; *TUG1*, Taurine-Upregulated 1; *XIST*, X Inactive Specific Transcript; *HCG18*, Major Histocompatibility Complex Group 18; *DICER1*, Dicer 1, Ribonuclease III; *E2F3*, E2F Transcription Factor 3; FZD6, Frizzled Class Receptor 6; *HIF1A*, Hypoxia-Inducible Factor 1 Subunit Alpha; *MYCN*, MYCN Proto-Oncogene, BHLH Transcription Factor; *REL*, REL Proto-Oncogene, NF-KB Subunit; *SNHG1*, Small Nucleolar RNA Host Gene 1; *VEGFA*, Vascular Endothelial Growth Factor A; *CDK6*, Cyclin-Dependent Kinase 6; *SIRT1*, Sirtuin 1; *EGFR*, Epidermal Growth Factor Receptor; *PSMA3-AS1*, Proteasome 20S Subunit Alpha 3 Antisense RNA 1; *OIP5-AS1*, Opa-Interacting Protein 5 Antisense RNA 1; *RNF138*, Ring Finger Protein 138; *CCND1*, Cyclin D1; *CCND2*, Cyclin D2; *HNRNPM*, Heterogeneous Nuclear Ribonucleoprotein M; *SERBP1*, SERPINE1 MRNA-Binding Protein 1; *EEF1A1*, Eukaryotic Translation Elongation Factor 1 Alpha 1; *ATP2A2*, ATPase Sarcoplasmic/Endoplasmic Reticulum Ca^2+^ Transporting 2; *CREG1*, Cellular Repressor Of E1A Stimulated Genes 1; *KLHDC10*, Kelch Domain-Containing 10; *MLXIP*, MLX-Interacting Protein; *PPTC7*, Protein Phosphatase Targeting COQ7; *PSMD7*, Proteasome 26S Subunit, Non-ATPase 7; *SRPR*, SRP Receptor; *SRPRA*, SRP Receptor Subunit Alpha.

**Table 2 genes-16-00791-t002:** Genes and their responsible circRNAs: establishing the circRNA–gene axis.

Gene ID	circRNA
**Upregulated**	
*GTF2I*	hsa_circ_0006944
*AMMECR1L*	hsa_circ_0005892
*GTF2H2C*	hsa_circ_0004914
*EXTL3*	hsa_circ_0003885
*GPAM*	-
*DNAJC10*	hsa_circ_0057256
*CUL3*	hsa_circ_0008309
*GLUD1*	hsa_circ_0019034
*CTC1*	hsa_circ_0008041
*ATP1A1*	hsa_circ_0013692
*RCBTB1*	hsa_circ_0030262
*DDX3X*	hsa_circ_0090290
*CTDSPL*	hsa_circ_0064843
*AP2B1*	hsa_circ_0043121
**Downregulated**	
*ATM*	hsa_circ_0007694
*AFF4*	hsa_circ_0007945
*AP3S2*	hsa_circ_0009156
*ADAM15*	hsa_circ_0014480
*ABR*	hsa_circ_0041188

Differentially expressed miRNAs in antral fluid collected from ovarian follicular cysts versus mature ovarian follicles in Holstein dairy cows. *GTF2I*, General Transcription Factor IIi; *AMMECR1L*, AMMECR Nuclear Protein 1-Like; *GTF2H2C*, General Transcription Factor IIH Subunit 2 Family Member C; *EXTL3*, Exostosin-Like Glycosyltransferase 3; *GPAM*, Glycerol-3-Phosphate Acyltransferase, Mitochondria; *DNAJC10*, DnaJ Heat Shock Protein Family (Hsp40) Member C10; *CUL3*, Cullin 3; *GLUD1*, Glutamate Dehydrogenase 1; *CTC1*, CST Telomere Replication Complex Component 1; *ATP1A1*, ATPase Na+/K+ Transporting Subunit Alpha 1; *RCBTB1*, RCC1- and BTB Domain-Containing Protein 1; *DDX3X*, DEAD-Box Helicase 3 X-Linked; *CTDSPL*, CTD Small Phosphatase-Like; *AP2B1*, Adaptor-Related Protein Complex 2 Subunit Beta 1; *ATM*, ATM Serine/Threonine Kinase; *AFF4*, ALF Transcription Elongation Factor 4; *AP3S2*, Adaptor-Related Protein Complex 3 Subunit Sigma 2; *ADAM15*, ADAM Metallopeptidase Domain 15; *ABR*, ABR Activator of RhoGEF and GTPase.

**Table 3 genes-16-00791-t003:** Top 10 predicted snRNAs with high degree and betweenness centrality.

snRNA	Degree	Betweenness
RN7SL585P	9	894.1123
RN7SKP91	11	827.2224
SNORD17	9	802.2572
RN7SKP22	8	777.9545
RNA5SP524	6	733.8872
RN7SKP232	8	727.4715
RNA5SP404	3	677.4443
RNA5SP29	3	601.4439
SNORD69	3	587.1188
RN7SKP8	8	581.8583

Predicted, differentially expressed miRNAs in antral fluid collected from ovarian follicular cysts versus mature ovarian follicles in Holstein dairy cows. RN7SL585P—RNA, 7SL, Cytoplasmic 585, Pseudogene; RN7SKP91—RN7SK Pseudogene 91; SNORD17—Small Nucleolar RNA, C/D Box 17; RN7SKP22—RN7SK Pseudogene 22; RNA5SP524—RNA, 5S Ribosomal Pseudogene 524; RN7SKP232—RN7SK Pseudogene 232; RNA5SP404—RNA, 5S Ribosomal Pseudogene 404; RNA5SP29—RNA, 5S Ribosomal Pseudogene 29; SNORD69—Small Nucleolar RNA, C/D Box 69; RN7SKP8—RN7SK Pseudogene 8.

**Table 4 genes-16-00791-t004:** Top 10 incRNAs with high degree and betweenness centrality.

lncRNA	Degree	Betweenness
KCNQ1OT1	507	50,840.21
NEAT1	504	48,867.59
XIST	454	39,551.45
OIP5-AS1	259	10,794.21
HCG18	256	10,484.17
MALAT1	249	9724.205
GABPB1-AS1	245	9648.505
HELLPAR	215	7300.995
TUG1	204	6706.267
MIR29B2CHG	171	4842.389

*KCQN1OT1*, KCNQ1 Opposite-Strand/Antisense Transcript 1; *XIST*, X Inactive Specific Transcript; *OIP5-AS1*, Opa-Interacting Protein 5 Antisense RNA 1; *HCG18*, Major Histocompatibility Complex Group 18; *MALAT1*, Metastasis-Associated Lung Adenocarcinoma Transcript 1; *GABPB1-AS1*, GA-Binding Protein Transcription Factor Subunit Beta 1, Antisense RNA 1; *HELLPAR*, HELLP-Associated Long Non-Coding RNA; *TUG1*, Taurine-Upregulated 1; *MIR29B2CHG*, MIR29B2 and MIR29C Host Gene.

**Table 5 genes-16-00791-t005:** Key miRNAs in OFCs: categorized by function.

miRNA	Function	Role in Ovarian Follicular Cysts	Functional Category
miR-26b	Promotes granulosa cell apoptosis via ATM, SMAD4, and HAS2 pathways	May impair follicle survival, promoting atresia and contributing to cyst persistence	Stress Response
miR-18a	Regulates TGF-β/SMAD signaling, induces apoptosis in granulosa cells	Dysregulation may disrupt follicular atresia or maturation, possibly leading to cystic changes	Stress Response
miR-30b/30e	Involved in granulosa cell proliferation, apoptosis, and autophagy	Altered expression may disturb follicular growth dynamics, favoring abnormal persistence	Stress Response
miR-15b	Regulates ovulation, follicular atresia, and steroidogenesis	Imbalance may impair ovulation timing, contributing to cyst formation	Steroidogenesis
miR-29a	Expressed in granulosa cells of mature dominant/subordinate follicles; ECM modulation	Dysregulation may influence dominance selection, leading to abnormal follicular development	ECM Remodeling
miR-191	Involved in immune signaling (e.g., Interleukin 6 (*IL6*), Toll-Like Receptor 3 (*TLR3*); proinflammatory	Changes in expression may impact the inflammatory milieu, affecting follicular stability	Stress Response/Insulin Resistance (NEB)
miR-132	Regulates hormone production, steroidogenesis, and NF-κB, *IL8* activity	Dysregulation may disturb hormonal balance and granulosa function, promoting cysts	Steroidogenesis/Stress Response
miR-221	Modulates ErbB, PI3K-Akt signaling, targets Fos Proto-Oncogene, AP-1 Transcription Factor Subunit (*FOS*), and Matrix Metallopeptidase 1 (*MMP1*)	Upregulation in subordinate follicles implies a role in follicular arrest and cystogenesis	Steroidogenesis/ECM Remodeling/Insulin Resistance (NEB)
miR-145	Controls follicle formation, maintenance, and activation	Downregulation in hyperstimulated follicles may impair normal follicle development	ECM Remodeling
miR-103	Involved in metabolic regulation; targets PLAG1 Zinc Finger (*PLAG1*), Cell Division Cycle-Associated 4 (*CDCA4*), Beta-Secretase 1 (*BACE1*)	Upregulation may alter granulosa cell metabolism and proliferation in hyperstimulated follicles	Negative Energy Balance/Insulin Resistance (NEB)
miR-101	Targets Cyclooxygenase 2 (*COX2*); modulates inflammation and prostaglandin production	Dysregulated inflammation may interfere with ovulation and promote cyst development	Stress Response
miR-193a	Present in both mature dominant and subordinate follicle libraries	Possible role in follicle fate; altered expression may favor cyst persistence	Steroidogenesis

Stress response could be cellular stress response (e.g., reactive oxygen species), ovarian stress response (e.g., hormone dysregulation), or cow stress response (e.g., nutrition, milk production).

**Table 6 genes-16-00791-t006:** Key miRNAs in PCOS: categorized by functional role.

miRNA	Primary Function	Role in PCOS	Functional Category
miR-18a	Regulates TGF-β/SMAD signaling, induces apoptosis	Overexpression promotes follicular persistence and anovulation	Stress Response/Cell Survival
miR-103	Influences insulin sensitivity; targets IRS pathway	Upregulated in PCOS; impairs insulin signaling in granulosa cells	Insulin Signaling
miR-221	Modulates ErbB, PI3K-Akt signaling, *MMP1*, *FOS*	Promotes insulin resistance, alters steroidogenesis, and affects follicular growth	Insulin Signaling/Cell Survival
miR-30b/30e	Cell proliferation, apoptosis, autophagy	Regulates granulosa survival and stress response	Stress Response/Cell survival
miR-21	Anti-apoptotic; inhibits caspase-3	Overexpression delays follicular atresia, promoting anovulation	Stress Response/Cell Survival
miR-132/320	Regulate steroidogenesis and granulosa cell function	Downregulation impairs estrogen synthesis, affecting follicle maturation	Steroidogenesis
miR-193a	Follicle fate regulation	Present in PCOS-related follicular environments	Cell Survival
miR-132	Regulates NF-κB, *IL8*, steroid hormone pathways	Downregulation impacts hormone synthesis and promotes abnormal follicular development	Steroidogenesis/Stress Response
miR-29a	Involved in extracellular matrix turnover and oocyte development	Dysregulation impairs follicle structure and ovulatory capacity	ECM Remodeling
miR-145	Regulates folliculogenesis and cell cycle checkpoints	Downregulation impairs granulosa cell differentiation and follicle progression	ECM Remodeling/Cell Cycle
miR-221	Modulates ErbB and PI3K-Akt signaling	Overexpression may block follicle maturation and arrest granulosa proliferation	Steroidogenesis/Insulin Signaling

Stress response could be cellular stress response (e.g., reactive oxygen species), ovarian stress response (e.g., hormone dysregulation), or human stress response (e.g., nutrition, milk production).

**Table 7 genes-16-00791-t007:** miRNA roles in bovine follicular cysts vs. human PCOS.

miRNA	Function	In Bovine Follicular Cysts	In Human PCOS
miR-26b	Promotes granulosa cell apoptosis via *ATM*, *SMAD4*, *HAS2*	Enhances atresia, reducing follicle survival, contributing to cyst formation	Not widely reported in PCOS
miR-21	Anti-apoptotic; inhibits caspase-3	Prevents apoptosis of granulosa cells in mature follicles	Overexpression promotes follicular persistence and anovulation
miR-18a	Regulates TGF-β/SMAD signaling, induces apoptosis	May disrupt follicular atresia and lead to cystic changes	Alters SMAD signaling; involved in follicular atresia
miR-30b/30e	Cell proliferation, apoptosis, autophagy	Dysregulation affects follicular dynamics and leads to abnormal persistence	Regulates granulosa survival and stress response
miR-15b	Ovulation, atresia, steroidogenesis	Impairs ovulation timing and contributes to cyst development	Associated with ovulatory dysfunction in PCOS
miR-29a	ECM remodeling and oocyte development	Dysregulation impairs mature dominant follicle selection	Disrupts follicular structure and ovulation
miR-132	Hormone production, NF-κB, *IL8* regulation	Disrupts steroid balance and granulosa cell function	Downregulated; contributes to low estrogen and inflammation
miR-221	Modulates Erb-B2 Receptor Tyrosine Kinase (*ErbB*), PI3K-Akt signaling, *MMP1*, *FOS*	Linked to follicular arrest in subordinate follicles	Promotes insulin resistance, alters steroidogenesis, and affects follicular growth
miR-145	Follicle maintenance and activation	Downregulated in hyperstimulated follicles, affects granulosa function	Downregulated in PCOS; impairs follicle activation
miR-103	Metabolic regulation, insulin signaling	Upregulation alters granulosa metabolism	Contributes to insulin resistance and metabolic dysfunction
miR-101	Targets *COX2*, inflammatory modulation	Disrupts ovulatory processes through inflammation	Inflammatory mediator; possible involvement in ovulatory failure
miR-193a	Follicle fate regulation	Associated with the persistence of subordinate follicles	Present in PCOS-related follicular environments

## Data Availability

The data presented in this study are available in the article or Appendix A herein.

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
