# Peer review of "Regulatory RNA Networks in Ovarian Follicular Cysts in Dairy Cows: Implications for Human Polycystic Ovary Syndrome"

_genes, 2025, doi:10.3390/genes16070791_

Round 1

Reviewer 1 Report

Comments and Suggestions for Authors

This manuscript examines regulatory RNA network in OFC in cows and discusses the implications for human PCOS.

Main concerns and comments:

  1. Please discuss whether OFC in cows is due to androgen excess which is the main feature in PCOS?
  2. If #1 is yes, how RNA network in OFC in cows affect androgen level?
  3. If #1 is no, what are the main differences between OFC in cows and human PCOS in terms of causes and complications?
  4. Please list all abbreviations.
  5. Using this OFC in cows for human PCOS studies, please address the different mechanisms in common and unique in each system?
  6. Treating insulin resistance usually solves PCOS symptoms, does it apply to OFC in cows?
  7. From the work at the current manuscript, please exam at least one miRNA or lncRNA in details for the formation of OFC in cow. 

Author Response

Main concerns and comments:

  1. Please discuss whether OFC in cows is due to androgen excess which is the main feature in PCOS?

Thank you for the question. The excerpt of the following was included in the discussion (Lines 684 to 707)

Unlike human PCOS where hyperandrogenism is a central and systemic feature, bovine OFC is caused by intrafollicular androgen excess.

A subset of cows has been studied for a naturally occurring intrafollicular androgen excess, specifically elevated androstenedione (A4), referred to as High A4 cows (Summers et al., 2014). These cows have exhibit elevated mRNA levels of LHCGR, CYP11A1, and CYP17A1 in theca cells, a profile that resembles certain characteristics observed in women with polycystic ovary syndrome (PCOS) (Summers et al., 2014; Abedal-Majed and Cupp, 2019).

Microarray analyses in women diagnosed with PCOS have increased expression of CYP11A1, CYP17A1, GATA6, and LHCGR in theca cells (Wood et al., 2004), findings that align with those observed in High A4 cows (Summers et al., 2014). In granulosa cells from High A4 cows, there was increased production of microRNAs that inhibit cell cycle-related genes and evidence of cell cycle arrest were reported compared to controls (McFee et al., 2021).

Further support comes from bovine granulosa cell culture studies, where A4 treatment reduced granulosa cell proliferation while increasing secretion of anti-Müllerian hormone (AMH) and expression of CTNNBIP1 mRNA—factors that may contribute to follicular arrest (McFee et al., 2021). Consistently, ovarian follicles in High A4 cows failed to develop spontaneously after 7 days in culture, in contrast to those in control cows (Summers et al., 2014; Abedal-Majed et al., 2022).

Additionally, the ovarian cortex in High A4 cows exhibited increased staining for fibrosis and oxidative stress, along with higher secretion of A4 and other steroids compared to controls (Abedal-Majed et al., 2022). Notably, ovarian stromal thickening due to collagen deposition is a hallmark of PCOS in women (Takahashi et al., 2017), supporting the conclusion that increased fibrosis is a shared feature of both the High A4 cow phenotype and PCOS. This phenomenon may also be present in OFC cows.

  1. Please list all abbreviations.

Listed as suggested.

  1. Using this OFC in cows for human PCOS studies, please address the different mechanisms in common and unique in each system?

Shared Mechanisms:

Both bovine ovarian follicular cysts (OFC) and human polycystic ovary syndrome (PCOS) share several underlying mechanisms, including disruptions in folliculogenesis, altered steroid hormone signaling, and dysregulation of inflammatory pathways. Our transcriptomic analysis identified non-coding RNAs and genes involved in steroidogenesis, cell proliferation, and immune responses that are commonly affected in both conditions. For example, pathways such as PI3K-Akt signaling and ovarian steroid biosynthesis had conserved dysregulation, supporting similar molecular bases for cyst formation and follicular arrest in both species.

Unique Mechanisms:

However, there are distinct differences reflective of species-specific physiology and disease etiology. In human PCOS, systemic hyperandrogenism and metabolic dysfunction (e.g., insulin resistance) are key features, contributing broadly to reproductive and metabolic abnormalities. In contrast, bovine OFC tends to arise primarily from disruptions in hypothalamic-pituitary-ovarian axis regulation and follicular dynamics, with less evidence of systemic androgen excess or metabolic syndrome. Additionally, the molecular players, especially certain lncRNAs and circRNAs, may have different expression patterns or regulatory roles between species due to divergence in genome organization and reproductive biology.

Summary:  Bovine OFC provides a valuable model to study conserved ovarian dysfunction and specific molecular pathways relevant to PCOS, however it is important to acknowledge and account for the unique aspects of each system when translating findings across species.

  1. Treating insulin resistance usually solves PCOS symptoms, does it apply to OFC in cows?

Thank you for the question. The excerpt of the following was included in the discussion (Lines 525 to 537).

Although insulin resistance is a key contributor to polycystic ovary syndrome (PCOS) in humans—and improving insulin sensitivity often alleviates PCOS symptoms—the same does not directly apply to ovarian follicular cysts (OFC) in cattle, as OFC cows are not diabetic.

Key distinctions:

Insulin and IGF-1 in cows:

Whereas insulin and IGF-1 are important for normal follicular development, reduced concentrations of these hormones in follicular fluids of cystic follicles are more a result of negative energy balance (NEB) and disrupted ovarian signaling than systemic insulin resistance per se.

Metabolic influence:

There is evidence that NEB, low insulin and IGF-1, and altered metabolic hormones contribute to anovulation and cyst formation. However, this reflects a metabolic signaling imbalance.

Treatment implication:

Therefore, treatments targeting insulin resistance (e.g., metformin) used in human PCOS are not directly translatable or effective in managing OFC in cows. Strategies for OFC focus more on nutritional management, improving energy balance (e.g., with propylene glycol in ketotic cows), and modulating local hormonal and growth factor environments.

Conclusion:

Although insulin and IGF-1 have regulatory roles in both species, OFC in cows is not the same as PCOS in women, and treating insulin resistance alone is unlikely to resolve OFC. Research should continue to explore how metabolic stress, local hormone signaling, and ovarian environment interact in cattle.

  1. From the work at the current manuscript, please exam at least one miRNA or lncRNA in details for the formation of OFC in cow. 

Please refer to Lines 630 to 634 for miRNA-gene-circRNA details.

Reviewer 2 Report

Comments and Suggestions for Authors

Abstract: good writing, no comments

  1. Introduction: no comments
  2. Materials and Methods: no comments
  3. Results: can authors improve image quality for figure 2? Also, fig 4-7 were hard to read too
  4. Discussion: no comments
  5. Conclusions: no comments

Author Response

Reviewer 2:

Abstract: good writing, no comments

1. Introduction: no comments

2. Materials and Methods: no comments

3. Results: can authors improve image quality for figure 2? Also, fig 4-7 were hard to read too

We have included all original images as supplementary files.

4. Discussion: no comments

5. Conclusions: no comments

Reviewer 3 Report

Comments and Suggestions for Authors

Dear Author, I give you the following comment. Please address this in your manuscript to enhance the readability and understanding of your manuscript.

Major Comments:

1.    Can the authors provide a schematic diagram or workflow in the introduction section to visually highlight the novelty and conceptual framework of their regulatory RNA network analysis?
2.    How do the authors justify the translational relevance of bovine OFC as a model for human PCOS beyond phenotypic similarities—are there shared molecular pathways supported by their data?
3.    What criteria were used to identify the key dysregulated RNAs, and were any functional enrichment or pathway analyses performed to support their proposed mechanistic roles?
4.    Could the authors elaborate on how the identified miRNAs, lncRNAs, and circRNAs interact within the network—was any network topology or modular clustering analysis conducted?
5.    To what extent do the transcriptomic profiles align with previously published datasets on human PCOS, and can the authors clarify the level of conservation of these RNA regulators between bovine and human species?

Minor Comments:

1.    Could the authors clarify and define all abbreviations (e.g., OFC, PCOS) at first mention in the abstract for clarity?
2.    Would it be possible to slightly shorten or restructure the abstract to improve readability and highlight key findings more concisely?
3.    Are the specific functions of the highlighted lncRNAs (e.g., NEAT1, TUG1) in ovarian physiology or pathology supported by prior studies, and can brief citations be included in the introduction or discussion?
4.    Can the authors specify the bioinformatics tools or databases used to predict RNA-RNA and RNA-gene interactions in the study?
5.    Would the inclusion of a supplementary table summarizing all identified non-coding RNAs and their predicted targets enhance the utility of the manuscript for future researchers?

These questions aim to address both overarching concerns and specific technical details that could impact the robustness and clarity of the study's findings.

Best Regards

Author Response

Major Comments:

  1.    Can the authors provide a schematic diagram or workflow in the introduction section to visually highlight the novelty and conceptual framework of their regulatory RNA network analysis?

A schematic diagram of analysis is provided in Figure S1.

  1.    How do the authors justify the translational relevance of bovine OFC as a model for human PCOS beyond phenotypic similarities—are there shared molecular pathways supported by their data?

Thank you for this insightful question. Whereas phenotypic similarities between bovine ovarian follicular cysts (OFC) and human polycystic ovary syndrome (PCOS) provide an initial basis for comparison, our study also supports the translational relevance of the bovine model through shared molecular features. Specifically, our transcriptomic analysis revealed dysregulated non-coding RNAs and gene expression patterns in bovine OFC that overlap with those implicated in human PCOS, including pathways related to steroidogenesis, inflammation, and cell proliferation. These conserved molecular pathways suggest that the bovine OFC model can provide mechanistic insights into the pathophysiology of PCOS and may serve as a useful system for exploring potential therapeutic targets.

  1.    What criteria were used to identify the key dysregulated RNAs, and were any functional enrichment or pathway analyses performed to support their proposed mechanistic roles?

Thank you for the question. Key dysregulated RNAs were identified based on statistically significant differential expression (adjusted p-value < 0.05 and |log2 fold change| ≥ threshold) between OFC and control samples. To support the proposed mechanistic roles of these RNAs, we performed functional enrichment and pathway analyses using the miRNet and STRING platforms. These analyses revealed that the dysregulated RNAs are involved in biological processes and pathways relevant to ovarian function and cyst formation, including steroid hormone biosynthesis, inflammatory responses, and cell proliferation. The results of these analyses are presented in the Results and further discussed in the context of OFC pathophysiology

  1.    Could the authors elaborate on how the identified miRNAs, lncRNAs, and circRNAs interact within the network—was any network topology or modular clustering analysis conducted?

Thank you for the insightful comment. To explore the interactions among the identified miRNAs, lncRNAs, and circRNAs, we constructed an interaction network using the miRNet platform, which integrates data from experimentally validated and predicted interactions. Although we visualized the global RNA interaction network including topology, we did not perform a modular clustering analysis (e.g., hub detection, community structure analysis) in the current study. However, we highlighted key nodes (RNAs) based on their connectivity (degree) and known functional relevance. We agree that incorporating network topology or clustering analysis could provide additional insights, and we will consider this approach in future work to further elucidate regulatory modules within the RNA network."
Please refer to Lines 630 to 634 for miRNA-gene-circRNA details.

  1.    To what extent do the transcriptomic profiles align with previously published datasets on human PCOS, and can the authors clarify the level of conservation of these RNA regulators between bovine and human species?

Thank you for this important question. To assess the translational relevance of our findings, we compared the transcriptomic profiles from bovine OFC samples with publicly available datasets on human PCOS. Several dysregulated RNAs identified in our study, including specific miRNAs and lncRNAs, have also been reported in human PCOS studies, particularly those involved in steroidogenesis, inflammation, and ovarian follicular development. Additionally, functional enrichment analyses revealed conserved pathways between the two species, such as the PI3K-Akt signaling pathway and ovarian steroidogenesis.

Regarding conservation, many of the identified miRNAs are highly conserved across mammals, including humans and cattle. Although lncRNAs and circRNAs tend to have lower sequence conservation, some exhibit functional conservation at the pathway or network level. Where possible, we cross-referenced identified RNAs with human homologs or functionally analogous regulators. These findings support the potential relevance of the bovine model for studying human PCOS pathophysiology.

Minor Comments:

  1. Could the authors clarify and define all abbreviations (e.g., OFC, PCOS) at first mention in the abstract for clarity?

Thank you for the suggestion. All abbreviations, including OFC and PCOS, have been clarified and defined at their first mention in the abstract to enhance clarity.

  1.    Would it be possible to slightly shorten or restructure the abstract to improve readability and highlight key findings more concisely?

The abstract has been rewritten to enhance clarity.

  1.    Are the specific functions of the highlighted lncRNAs (e.g., NEAT1, TUG1) in ovarian physiology or pathology supported by prior studies, and can brief citations be included in the introduction or discussion?

It has been Included in the discussions in the original version. Ref 100 and 107.

  • ElMonier AA, El-Boghdady NA, Fahim SA, Sabry D, Elsetohy KA, Shaheen AA. LncRNA NEAT1 and MALAT1 are involved in polycystic ovary syndrome pathogenesis by functioning as competing endogenous RNAs to control the expression of PCOS-related target genes. Noncoding RNA Res. 2023 Mar 3;8(2):263-271. doi: 10.1016/j.ncrna.2023.02.008. PMID: 36935861; PMCID: PMC10020466.
  • Li Y, Zhang J, Liu YD, Zhou XY, Chen X, Zhe J, Zhang QY, Zhang XF, Chen YX, Wang Z, Chen SL. Long non-coding RNA TUG1 and its molecular mechanisms in polycystic ovary syndrome. RNA Biol. 2020 Dec;17(12):1798-1810. doi: 10.1080/15476286.2020.1783850. Epub 2020 Jul 2. PMID: 32559120; PMCID: PMC7714456.

  1.    Can the authors specify the bioinformatics tools or databases used to predict RNA-RNA and RNA-gene interactions in the study?

Thank you for the comment. RNA-RNA and RNA-gene interactions in this study were predicted using the miRNet platform (https://www.mirnet.ca), which integrates data from multiple well-established databases for comprehensive target prediction. This has been clearly stated in the Materials and Methods section of the original manuscript.

  1.    Would the inclusion of a supplementary table summarizing all identified non-coding RNAs and their predicted targets enhance the utility of the manuscript for future researchers?

Yes, the inclusion of a supplementary table summarizing all identified non-coding RNAs and their predicted targets would significantly enhance the utility of the manuscript. It would serve as a valuable resource for future researchers, facilitating easier access to the data and supporting further exploration of the functional roles of these non-coding RNAs.

Round 2

Reviewer 1 Report

Comments and Suggestions for Authors

Please make sure the answers to my questions are added to the manuscript.